# Bayesian latent class models for identifying canine visceral leishmaniosis using diagnostic tests in the absence of a gold standard

**Marie V. Ozanne**[1]*, **Grant D. Brown**[2], **Breanna M. Scorza**[3], **Kurayi Mahachi**[3], **Angela J. Toepp**[4,5], **Christine A. Petersen**[3,6]

**1** Department of Mathematics & Statistics, Mount Holyoke College, South Hadley, Massachusetts, United States of America, **2** Department of Biostatistics, University of Iowa College of Public Health, Iowa City, Iowa, United States of America, **3** Department of Epidemiology, University of Iowa College of Public Health, Iowa City, Iowa, United States of America, **4** Enterprise Analytics, Sentara Healthcare, Virginia Beach, Virginia, United States of America, **5** Department of Internal Medicine, Eastern Virginia Medical School, Norfolk, Virginia, United States of America, **6** Center for Emerging Infectious Diseases, University of Iowa College of Public Health, Iowa City, Iowa, United States of America

* mozanne@mtholyoke.edu

**Data Availability Statement:** The data underlying the results presented in the study are available on

## Abstract

### Background

Like many infectious diseases, there is no practical gold standard for diagnosing clinical visceral leishmaniasis (VL). Latent class modeling has been proposed to estimate a latent gold standard for identifying disease. These proposed models for VL have leveraged information from diagnostic tests with dichotomous serological and PCR assays, but have not employed continuous diagnostic test information.

### Methods/Principal findings

In this paper, we employ Bayesian latent class models to improve the identification of canine visceral leishmaniasis using the dichotomous PCR assay and the Dual Path Platform (DPP) serology test. The DPP test has historically been used as a dichotomous assay, but can also yield numerical information via the DPP reader. Using data collected from a cohort of hunting dogs across the United States, which were identified as having either negative or symptomatic disease, we evaluate the impact of including numerical DPP reader information as a proxy for immune response. We find that inclusion of DPP reader information allows us to illustrate changes in immune response as a function of age.

### Conclusions/Significance

Utilization of continuous DPP reader information can improve the correct discrimination between individuals that are negative for disease and those with clinical VL. These models provide a promising avenue for diagnostic testing in contexts with multiple, imperfect diagnostic tests. Specifically, they can easily be applied to human visceral leishmaniasis when diagnostic test results are available. Also, appropriate diagnosis of canine visceral leishmaniasis has important consequences for curtailing spread of disease to humans.

ResearchGate (https://www.researchgate.net/project/Leishmania-detection).

**Funding:** CAP and GDB were supported under Fogarty International Center of the National Institutes of Health, Award Number R01TW010500 (https://www.fic.nih.gov/). BMS was supported under National Institutes of Health, National Institute of Allergy and Infectious Diseases; Award Number T32AI007260 (https://www.niaid.nih.gov/). The funders had no role in the study design, data collection and analysis, decision to publish, or preparation of the manuscript.

**Competing interests:** The authors have declared that no competing interests exist.

## Author summary

Diagnostic tests often aid in diagnosis of infectious diseases, like the neglected tropical disease visceral leishmaniasis (VL). While such tests provide useful information, they are imperfect, so they can produce false positive or false negative results. Moreover, diagnostic tests for VL are often dichotomous, meaning they only provide positive or negative results. The Dual Path Platform (DPP) serology test has been used as a dichotomous assay, but can also yield numerical information via the DPP reader. Although we can only observe diagnostic test results, rather than the true disease status of an individual, we can use a set of statistical methods called latent class models to estimate the true disease status. We can also incorporate the numerical information from the DPP reader in latent class models. In this paper, we evaluate the impact of including this numerical information in a Bayesian latent class model to provide additional information about immune response. We compare this to a more traditional latent class model, which only incorporates dichotomized assay information. We fit these models to data collected from a cohort of hunting dogs across the United States. Incorporating DPP reader information allows us to illustrate changes in immune response for different ages.

## Introduction

Leishmaniasis is a neglected tropical disease that is endemic in 98 countries and three territories. The most severe manifestation of leishmanaisis is visceral (VL); it is fatal in 10% of human cases. Six countries, India, Bangladesh, Sudan, South Sudan, Ethiopia, and Brazil, account for more than 90% of human VL cases [1]. It also presents a serious risk to domesticated dogs, with a prevalence between two percent and 33 percent, depending on location [2]. In the Americas, this infection is caused by the parasite *Leishmania infantum* (*L. infantum*) and is zoonotic; dogs are recognized as the main animal reservoir [3]. In this capacity, canine visceral leishmaniasis (CVL) is a significant risk factor for human disease, which makes disease identification in dogs critical to public health [4–6].

Diagnostic tools for identifying CVL include parasite culture, serology, PCR, and clinical assessment of symptoms by licensed veterinarians. As noted by Solcà et al., the reliability of serological and PCR test results in particular depends heavily on whether the sample is extracted from the site of infection [7]. Parasite culture, which is often considered the "gold standard" for CVL diagnosis [8, 9], has low sensitivity for individuals that have a low parasite load [10, 11]. Clinical assessment relies on the feasibility of performing physical examination and is subject to expert variability.

Latent class analysis (LCA) represents a statistical solution to the problem of imperfect or infeasible diagnostic tools. Indeed, LCA has been used to evaluate the accuracies of various dichotomized diagnostic tests, including quantitative PCR and serology for CVL, as well as hematological parameters [7, 12, 13]. In this paper, we compare two Bayesian latent class models based on the Dual Path Platform (DPP) serology test and PCR. The DPP test, while historically used as a dichotomous assay, also can yield numerical test information from the DPP reader. The aim of this study is to evaluate the utility of including numerical DPP reader information, as a proxy for strength of immune response, in a latent class model. This is carried out for symptomatic and negative dogs as defined below, and is compared to a traditional approach, which incorporates dichotomized test information in identifying an underlying disease state. While LCA models are applied to CVL in this paper, these methods are sufficiently general to be easily applied to human clinical VL identification.

## Materials and methods

### Ethics statement

All dog caretakers gave signed informed consent, following a protocol approved by the University of Iowa Institutional Animal Care and Use Committee (IACUC).

### Study population

A cohort of hunting dogs from multiple locations across the United States had whole blood and serum collected. These dogs were naturally infected by *L. infantum* transplacentally. Physical exams were performed by veterinarians after the blood was collected. Demographic characteristics are summarized in Table 1. Data are available on ResearchGate: https://www.researchgate.net/project/Leishmania-detection.

### Clinical status

Dogs were defined as negative for *L. infantum* infection if they had both a negative qPCR and a negative DPP CVL test result. Dogs were defined as asymptomatic if they were positive on at least one diagnostic test (qPCR, DPP CVL) and presented with < 2 clinical signs. Dogs were defined as symptomatic if they had ≥ 2 clinical signs and were positive on one or both diagnostic tests (qPCR, DPP CVL). Clinical signs include: lymphadenopathy, cachexia, weight loss, dermatitis, alopecia, poor hair coat, conjunctivitis, epistaxsis, and splenomegaly. Only dogs that were defined as negative or symptomatic, based on these criteria, were included in this analysis. Asymptomatic dogs have a large range of immune responses and ability to develop antibodies to *Leishmania* parasites, which makes DPP reader scores highly variable and unreliable, making them very difficult to include in this kind of study. For the purposes of this research they were excluded.

### Diagnostic tests

DNA isolated from whole blood using the QIAamp DNA Blood Mini Kit (Qiagen) was analyzed for presence of *Leishmania* ribosomal DNA via Real Time-quantitative PCR (RT-qPCR) using the sequences: *forward* 5'-AAGTGCTTTCCCATCGCAACT-3'; *reverse* 5'-CGCACTA AACCCCTCCAA-3'; probe 5'-FAM-CGGTTCGGTGTGTGGCGCC-MGB NFQ-3' (ThermoFisher) as previously described [14].

Canine serum was used to perform the DPP CVL serological test (ChemBios) according to manufacturer instructions with detection via Chembios Diagnostic Systems, Inc., (Medford, NY) DPP digital Micro Reader. Micro Reader results < 10 are considered negative; results ≥ 10 are considered positive. The DPP CVL detects antibodies using recombinant rK28 antigen derived from *L. infantum* [15].

**Table 1. Number of observations, sex (% male), median age (0.025 and 0.975 quantiles) in years, and median DPP reader (0.025 and 0.975 quantiles), summarized by dichotomous assay test results.**

| Test result | n | Sex (% Male) | Age (0.025, 0.975) | DPP (0.025, 0.975) |
|---|---|---|---|---|
| DPP (+), PCR (+) | 73 | 65.8 | 4.3 (2, 8) | 229.0 (20.0, 337.0) |
| DPP (+), PCR (−) | 148 | 47.3 | 4.5 (1, 10) | 67.7 (10.0, 230.0) |
| DPP (−), PCR (+) | 11 | 81.8 | 3.2 (1, 8.2) | 2.69 (0.9, 5.9) |
| DPP (−), PCR (−) | 1077 | 48.9 | 3.8 (0.7, 9) | 2.8 (0.6, 8.1) |

## Statistical methods

In this paper, we introduce Bayesian latent class hierarchical models to estimate the underlying disease state for each observation based on properties and results from two diagnostic tests, as discussed above. Further, demographic variables age, sex, an indicator for senior (6 years or older), and an interaction between senior and age were included in the model based on their prior association with leishmaniasis and disease progression [4, 16, 17]. Of the 1309 observations included in the analysis, 307 are classified as senior. We assume that all observations are independent. We also assume that the two diagnostic tests are conditionally independent. This is common practice in latent class modeling for this type of application; it is a valid assumption here because the biological mechanisms on which the tests rely are different. The DPP test relies on antibody production by the canine immune system, indicating previous exposure to the parasite, which may or may not be coincident with concurrent infection, although rising rK28 antibody levels are associated with disease progression [18]. In contrast, a positive qPCR result is indicative of current infection by *Leishmania* parasites at a threshold detectable level in the bloodstream, and is independent of the humoral response of the host.

**Notation.**   For $i \in 1, \ldots, n$, where $n$ is the total number of observations, and diagnostic test $j \in 1, 2$:

- $Y_{ij}$: test $j$ outcome for observation $i$;

- $\eta_j$: sensitivity of test $j$;

- $\gamma_j$: specificity of test $j$;

- $D_i$: underlying disease status for observation $i$;

- $\boldsymbol{\theta}$: linear effects for intercept, age, sex (male baseline), senior ($\geq$ 6 years), and an age/senior interaction

**Data model.**   While our approach can be easily generalized to more complex scenarios, here we consider two diagnostic tests. The first, DPP, can be either dichotomized or continuous, due to the presence of the DPP reader. If the test is used as a dichotomous assay, then

$$Y_{ij}|D_i \sim \text{Bernoulli}(\delta_{ij}), \tag{1}$$

where $\delta_{ij} = D_i \, \eta_j + (1 - D_i)(1 - \gamma_j)$. Alternatively, if it is being treated as continuous, then we model it using a mixture of gamma distributions,

$$Y_{ij}|D_i \sim \pi_i g_1(y_{ij}) + (1 - \pi_i)g_0(y_{ij}) \tag{2}$$

where $g_1(\cdot)$ is the density function corresponding to a $\Gamma(\alpha_1, \beta_1)$ and $g_0(\cdot)$ is the density function corresponding to a $\Gamma(\alpha_0, \beta_0)$ distribution. Discussion of how these distribution parameters are determined follows in the Prior Model and Hyperparameters section.

**Process model.**   The latent disease state for individual $i$, $D_i$, is distributed

$$D_i \sim Bernoulli(\pi_i), \tag{3}$$

where

$$\pi_i = \text{logit}^{-1}(\mathbf{x}_i^T \boldsymbol{\theta}); \tag{4}$$

$\boldsymbol{\theta} = (\theta_0, \theta_1, \theta_2, \theta_3, \theta_4)$, where $\theta_0$ is related to the logit of the prevalence of the disease in the population, although the interpretation is complex; $\theta_1$ and $\theta_2$ are the effects of sex (baseline is male) and age (in years), respectively; $\theta_3$ is the effect of senior; $\theta_4$ is the effect of the interaction between senior and age. The design matrix, $\mathbf{x}_i^T$, is a $1 \times 5$ matrix:

$[1 \quad I_{\text{Female}} \quad a \quad I_{\text{Senior}} \quad a \times I_{\text{Senior}}]$; $I_{\text{Female}}$ is 1 if individual $i$ is female and 0 otherwise, $I_{\text{Senior}}$ is 1 if individual $i$ is older than 5 and 0 otherwise, and $a$ is the age, in years, of individual $i$.

**Prior model and hyperparameters.** Informative priors are placed on $\eta_j$, $\gamma_j$, $\alpha_0$, $\alpha_1$, $\beta_0$, and $\beta_1$. The first two priors are $\eta_j \sim \text{Beta}(S_j/(1 - S_j), 1)$ and $\gamma_j \sim \text{Beta}(P_j/(1 - P_j), 1)$, where $S_j$ and $P_j$ are the mean sensitivity and specificity for test $j$, respectively. This approach allows researchers to utilize prior information on the sensitivity and specificity of various assays, which is generally available in the literature, while also not ignoring the remaining uncertainty in these terms.

The priors for the two gamma distributions, $\alpha_0$, $\alpha_1$, $\beta_0$, and $\beta_1$ are determined using past information about the mean and variance of the DPP score for negative and symptomatic dogs, respectively, as determined using clinical status. Note, clinical status is not explicitly included in the model in another capacity, and it is used to ensure identifiability for the mixture of gammas. For negative dogs, the average and standard deviation of the DPP score were $m_0 = 2$ and $s_0 = 1$, respectively. For symptomatic dogs, as defined in the methods section on clinical status, the average DPP reader score was $m_1 = 150$ and the standard deviation was $s_1 = 103$. Using these estimates, along with the analytical form of the mean and variance for gamma random variables, we established exponential hyperpriors on $g_0$ and $g_1$ with means equal to $m_0^2/s_0^2$ and $m_0/s_0^2$, and $m_1^2/s_1^2$ and $m_1/s_1^2$, respectively.

The prior distribution for the coefficients in Eq 4 was $\boldsymbol{\theta} \sim MVN(\boldsymbol{\mu}_\theta, \Sigma_\theta)$; the first element of $\boldsymbol{\mu}_\theta$, is selected to correspond to a prevalence of 0.1 [19]. It is standard practice in these models to use an informative prior for the prevalence.

## Simulation study

In this simulation study, we examine the relationship between sample size and prevalence, and the relationship with posterior predictive variability in the context of the latent class model which incorporates the DPP reader score through a mixture of gammas. To accomplish this, we simulate data with demographic properties similar to those in the real data employed here; we alter the population prevalence by varying the simulation value of $\theta_0$, which is related to disease prevalence for the average aged male. We fix $\theta_1$ through $\theta_4$ according to the effect sizes observed in the real data model fit. Then, we fit the simulated data sets according to the Statistical Methods section. Relevant files for conducting this simulation study are included in the S1 Appendix.

## Statistical software

Each of the models was fit using Markov chain Monte Carlo; this was implemented in the opensource software R using the `nimble` and `coda` packages [20–22]. Model convergence was assessed using the Gelman-Rubin diagnostic. The relevant files are available to the reader in the S1 Appendix.

## Results

The demographic variables, summarized by the test results, are recorded in Table 1. The majority of the observations corresponded to PCR and DPP negative results. The median ages were comparable across the test result combinations. The percent male was also comparable across the test results; there were only eleven dogs with a PCR positive and DPP negative result; nine were male.

**Table 2. Posterior mean and median estimates, 95% posterior credible intervals, and the posterior probability that a parameter is greater than 0 for Model 1, which uses dichotomized PCR and DPP test results, as well as sex, age in years, and an indicator for senior.** Senior is defined as 6 years or older (307 of 1309 observations); male is the baseline for sex. For parameters that are bounded above 0, $P(X > 0)$ is omitted. DPP = Dual Path Platform.

| Parameter | Description | Mean | Median | 2.5% | 97.5% | $P(X > 0)$ |
|---|---|---|---|---|---|---|
| $\theta_0$ | Intercept | -4.20 | -4.19 | -5.10 | -3.34 | 0.00 |
| $\theta_1$ | Sex=Female | -0.26 | -0.25 | -0.82 | 0.27 | 0.17 |
| $\theta_2$ | Age | 0.50 | 0.51 | 0.28 | 0.73 | 1.00 |
| $\theta_3$ | Senior | 0.56 | 0.57 | -1.13 | 2.23 | 0.74 |
| $\theta_4$ | Age/Senior Interaction | -0.39 | -0.39 | -0.67 | -0.12 | 0.00 |
| $\eta_1$ | DPP Sensitivity | 0.96 | 0.97 | 0.89 | 1.00 | – |
| $\eta_2$ | PCR Sensitivity | 0.74 | 0.73 | 0.51 | 0.98 | – |
| $\gamma_1$ | DPP Specificity | 0.97 | 0.97 | 0.95 | 0.99 | – |
| $\gamma_2$ | PCR Specificity | 1.00 | 1.00 | 0.99 | 1.00 | – |

## Model 1: Dichotomized PCR and DPP

The posterior mean and median estimates, and corresponding 95% credible intervals for the intercept, the effects of sex, age, senior, and the interaction between age and senior, as well as those for sensitivity and specificity for PCR and DPP are given in Table 2. The mean population prevalence for average aged males (3.9 years in this data set) is estimated at 9.5%, with a 95% credible interval of (1.8%, 37.9%). This interval is consistent with prevalence ranges reported for CVL [3, 19]. The mean odds of disease for males are 1.30 times those for females in this population, with a posterior 95% credible interval of (0.78, 2.27). There is an 83% chance that the mean odds of disease for males is greater than that for females in this population. This is consistent with past findings on the importance of considering sex as a risk factor for VL [23].

Fig 1 shows the median posterior predictive probability of disease as a function of age for each of the possible four combinations of PCR and DPP test results for Model 1. The probability of disease is clearly positively associated with age for all four test result combinations for those that are no more than 5 years old. In this age range, those that are DPP positive tend to have higher median probability of disease, although there is still considerable overlap in credible intervals among these four groups. In the canine population represented in this study, dogs that become infected with the *Leishmania* parasite tend to progress to clinical disease by age 5, after which they often succumb to disease. Survivors, represented by those that live with infection beyond age 5, tend to live to age 12, at which point they succumb to disease or die from other natural causes. The results from Model 1 are consistent with these observations.

## Model 2: Dichotomized PCR and DPP reader

Table 3 contains information on the posterior mean and median estimates, 95% posterior credible intervals, and posterior probability that a parameter is greater than 0 for Model 2, which differs from the model discussed in the previous section as it contains DPP reader information, rather than a dichotomized test result. Based on this model, the mean population prevalence for average aged males (3.9 years) is estimated at 17.2%, with a 95% credible interval of (5.4%, 42.4%). This interval is higher than that obtained from Model 1 but is consistent with findings in the United States in dog populations where leishmaniasis is endemic [3].

The median posterior predictive probability of disease as a function of age for Model 2 is shown in Fig 2 for each combination of PCR test result and DPP reader range (DPP < 10 corresponds to negative DPP, $10 \leq DPP < 100$ and $DPP \geq 100$ correspond to two potential levels

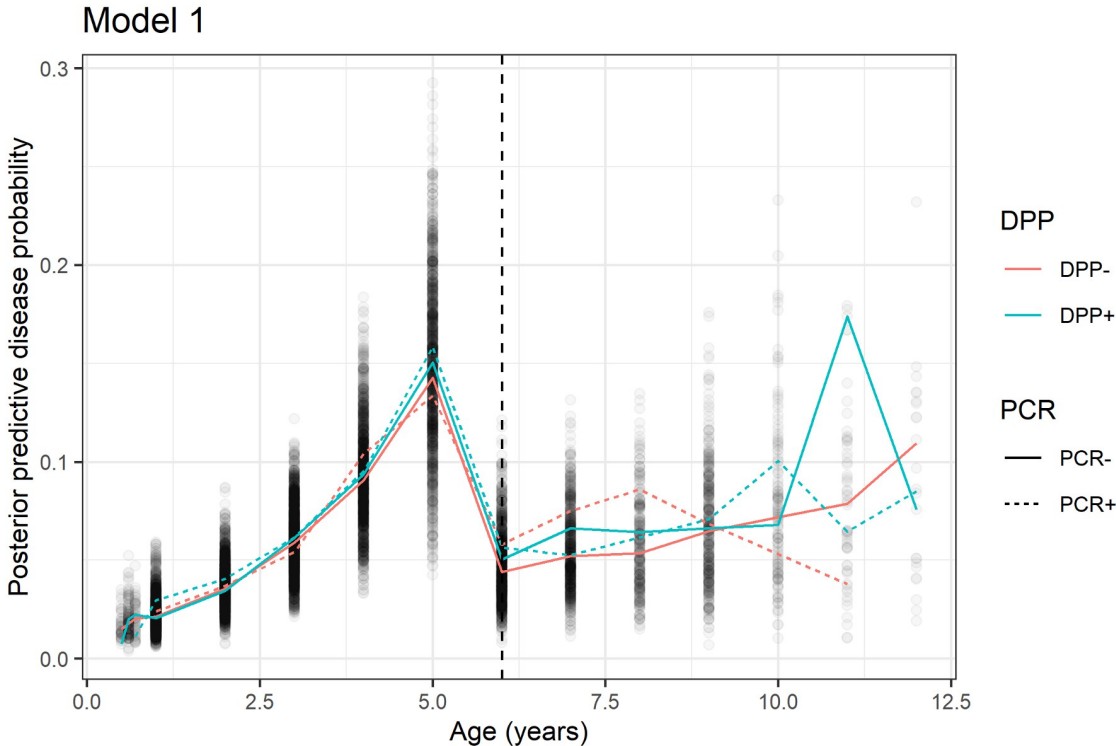

**Fig 1. Median posterior predictive probability of disease as a function of age for all four combinations of PCR and DPP test results.** We observe a peak in probability of disease at 5, after which dogs often die. Based on this model, those that survive past age 5 (307 of 1309 observations) never attain the same probability of disease. Due to the complex nature of this disease, there is still considerable uncertainty in these estimates; credible intervals are omitted from the graph.

of immune response). In this plot, we observe a similar trend to that from Model 1 for individuals less than 6 years old. Those that are PCR positive or have a DPP reader score greater than 100 have the highest median posterior predictive probability of disease at a recorded age of 5 years. Individuals that are PCR positive with a lower DPP reader score may be experiencing "rapid onset" of disease, which indicates that they are unable to control parasite replication and increases that likelihood of disease at an earlier age. Those that are PCR negative and DPP > 100 may be experiencing one of two immune responses that account for their test results. Individuals in this group may be "controllers", which are not as highly infected, as evidenced by the negative PCR test result but are controlling parasite replication through an immune response, which accounts for the high DPP reader score. Alternatively, individuals in this group may be highly infected with *Leishmania* and no longer mounting a strong immune response. This model cannot distinguish between these two scenarios. Those that are PCR negative with DPP > 100 that survive past age 5 attain a similar risk of disease by around age 9. Generally, those that survive past age 5 have a higher probability of disease if they have a positive PCR test. As in Model 1, the posterior credible intervals are still quite wide, so there is uncertainty in this model.

## Simulation study

The results from the simulation study for Model 2 are summarized in Fig 3. The uncertainty in posterior predictive probabilities clearly decreases for larger sample sizes for all included values

**Table 3. Posterior mean and median estimates, 95% posterior credible intervals, and the posterior probability that a parameter is greater than 0 for Model 2, which utilizes the DPP reader (continuous) and dichotomized PCR test results, as well as sex, age in years, and an indicator for senior.** Senior is defined as 6 years or older; male is the baseline for sex. For parameters that are bounded above 0, $P(X > 0)$ is omitted. DPP = Dual Path Platform.

| Parameter | Description | Mean | Median | 2.5% | 97.5% | $P(X > 0)$ |
|---|---|---|---|---|---|---|
| $\theta_0$ | Intercept | -3.17 | -3.16 | -3.84 | -2.57 | 0.00 |
| $\theta_1$ | Sex=Female | -0.19 | -0.19 | -0.58 | 0.19 | 0.17 |
| $\theta_2$ | Age | 0.41 | 0.41 | 0.25 | 0.58 | 1.00 |
| $\theta_3$ | Senior | -0.02 | -0.01 | -1.41 | 1.36 | 0.50 |
| $\theta_4$ | Age/Senior Interaction | -0.21 | -0.21 | -0.43 | 0.00 | 0.03 |
| $\eta_2$ | PCR Sensitivity | 0.39 | 0.39 | 0.30 | 0.48 | – |
| $\gamma_2$ | PCR Specificity | 1.00 | 1.00 | 0.99 | 1.00 | – |
| $\alpha_0$ | Shape: $f(Y_1\|D=0)$ | 3.37 | 3.36 | 3.02 | 3.74 | – |
| $\alpha_1$ | Shape: $f(Y_1\|D=1)$ | 0.62 | 0.61 | 0.49 | 0.78 | – |
| $\beta_0$ | Rate: $f(Y_1\|D=0)$ | 1.57 | 1.56 | 1.37 | 1.77 | – |
| $\beta_1$ | Rate: $f(Y_1\|D=1)$ | 0.01 | 0.01 | 0.00 | 0.01 | – |

of disease prevalence. Thus, the gamma mixture model approach performs better as sample size increases, which is expected.

## Discussion

In this paper, we employ Bayesian latent class modeling to improve the identification of clinical CVL. While this type of analysis has been performed before using a variety of dichotomized

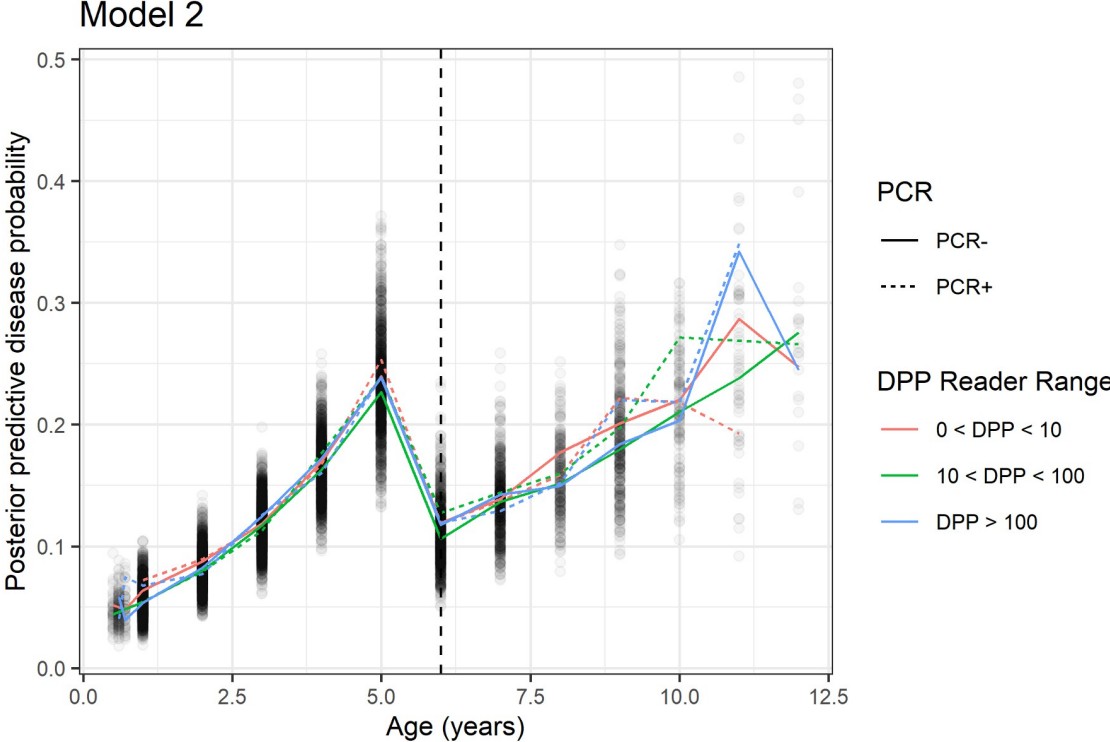

**Fig 2. Median posterior predictive probability of disease as a function of age for six combinations of PCR and DPP reader (interval) test results.** We observe a peak in probability of disease at 5, after which dogs often die. Based on this model, those that survive past age 5 attain the same probability of disease around age 10. Due to the complex nature of this disease, there is still considerable uncertainty in these estimates; credible intervals are omitted from the graph.

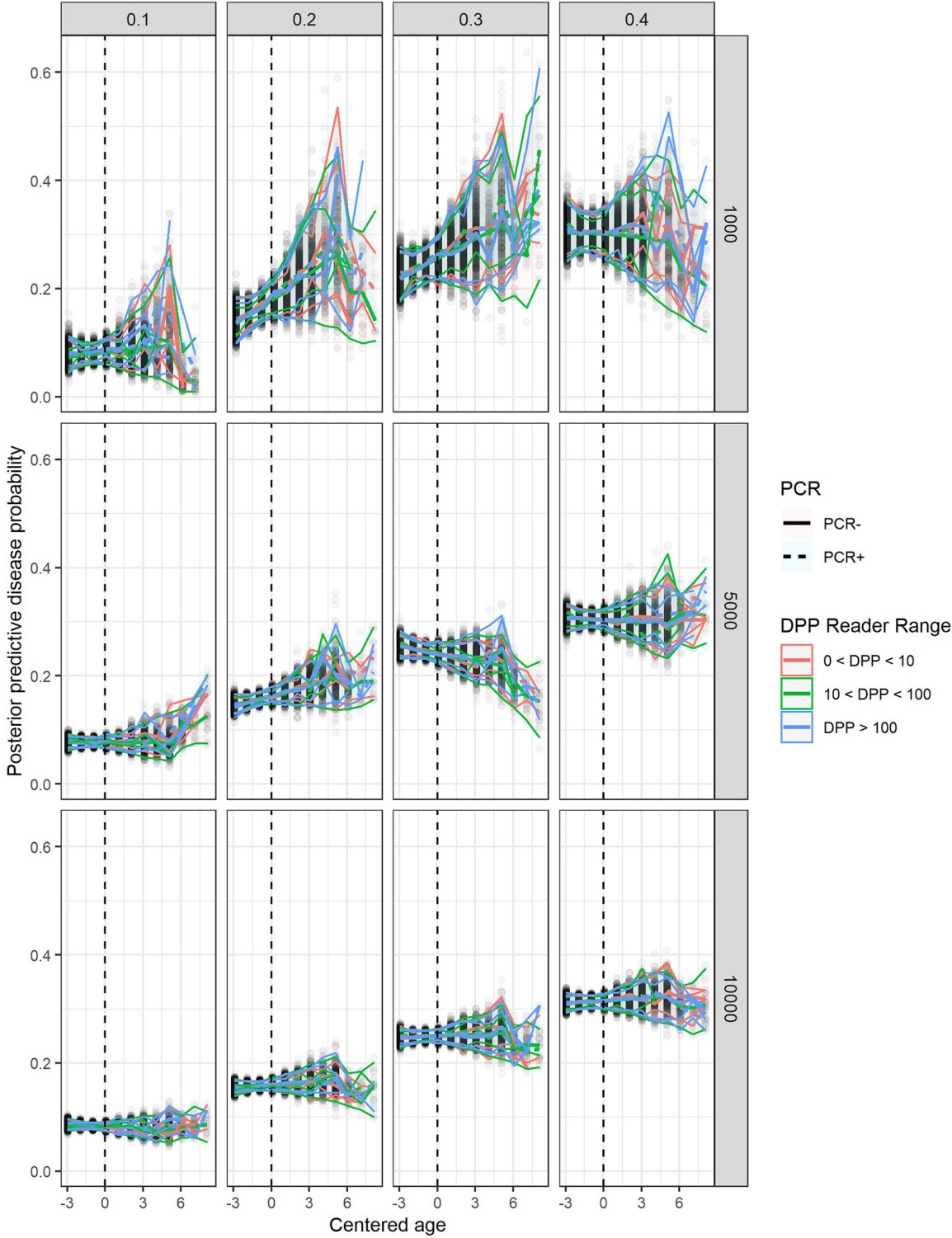

**Fig 3. Median posterior predictive probability (and variability, shaded) of disease as a function of age for six combinations of PCR and DPP reader (interval) test results for simulated data.** Prevalences (0.1–0.4; columns) and sample sizes (1000, 5000, 10000; rows) are varied. Posterior predictive variability decreases for larger sample sizes across all prevalence values.

diagnostic tests for leishmaniasis [7], in this paper we compare the utility of including numeric serology information from the DPP reader to that of including dichotomized serology test results. The modeling results presented in this paper suggest that inclusion of numeric DPP reader information, in the context of a gamma mixture model, can be helpful for illustrating changes in immune response as a function of age. Specifically, the DPP reader information allows us to reconcile conflicting dichotomized test results, particularly in cases where the DPP reader score is larger than 100 but the PCR test is negative.

These results were obtained in a sample of dogs that were classified either as negative or symptomatic for *L. infantum* based on clinical examination. While they play an important role in transmission, asymptomatics were excluded from this study since it was important to evaluate the model performance for the more distinct groups of negative and symptomatic dogs. Application of this latent class modeling approach that incorporates the DPP reader information for asymptomatics (based on clinical examination) is a future line of study. It is possible that the additional information afforded by the DPP reader could help address the challenging problem of identifying and treating asymptomatic individuals appropriately; however this will require further inquiry and is not a conclusion of this paper.

## Conclusion

While these models were developed in the context of a canine population, the methods, as well as possible extensions, are easily applied to human VL, where similar diagnostic tests and clinical examinations are utilized in disease diagnosis. In situations where clinical examination may not be immediately possible, but diagnostic testing is, models such as those presented in this paper may facilitate diagnosis, flagging individuals who are in need of further examination. It is also important to note that effectively diagnosing CVL has important consequences for curtailing human VL infection, since dogs serve as the primary animal reservoir [3]. Furthermore, studying VL in dogs can yield important information about human infections and disease, since parallels exist between infection and disease progression between human and their canine companions [24, 25].

There are several potential limitations of the analyses presented. First, most of the individuals included in these data had negative PCR and DPP results, which is reflected in the low average posterior probabilities of disease depicted in Fig 1. Also, the small number of observations with positive PCR test results may hamper the estimated PCR sensitivity in Model 2, in the presence of numeric DPP. As demonstrated in the simulation studies, the variability associated with the posterior predictive probabilities for the LCA with the gamma mixture model decreases for larger sample sizes. Interestingly, it does appear that including numeric DPP test results in addition to the dichotomous PCR assay may bolster identification of diseased individuals, even when the number of positive PCR tests in the sample is low. Second, the prior information for the mixture of gammas to model DPP reader score was based on clinical status, which was not included in the model in another capacity, from the same study. Hyperprior distributions were placed on the gamma parameters, but strictly speaking, prior information on these parameters should be taken from other prior studies, which were not available in this case. One alternative method of imposing ordering on the gamma distributions, $g_1(\cdot)$ and $g_0(\cdot)$, would be to introduce appropriate constraints concerning their means. Specifically, one could employ the constraint, $\alpha_1/\beta_1 > \alpha_0/\beta_0$, where $\alpha_1, \alpha_0, \beta_1, \beta_0$ are the shape and rate parameters for $g_1(\cdot)$ and $g_0(\cdot)$, respectively. Then, vague gamma priors could be placed on those shape and rate parameters. This took considerably more time to fit, however, and ran into convergence problems. In subsequent applications of these methods to other data sets, this prior information on

DPP score as a function of clinical status would be available, so this is not a limitation of the method. Rather, it is a limitation that exists only due to the relative novelty of the DPP reader.

## Supporting information

**S1 Appendix. R code for simulations and model fitting.** Separate files are included for each model implementation, as well as dependent functions.
(ZIP)

## Acknowledgments

We would like to thank the owners of the hunting dogs for allowing participation in this study and multiple veterinarians for helping us obtain the samples upon which this work is based.

## Author Contributions

**Conceptualization:** Marie V. Ozanne, Grant D. Brown.

**Data curation:** Breanna M. Scorza, Kurayi Mahachi, Angela J. Toepp, Christine A. Petersen.

**Formal analysis:** Marie V. Ozanne.

**Funding acquisition:** Grant D. Brown, Breanna M. Scorza, Christine A. Petersen.

**Investigation:** Marie V. Ozanne.

**Methodology:** Marie V. Ozanne.

**Supervision:** Christine A. Petersen.

**Visualization:** Marie V. Ozanne, Breanna M. Scorza.

**Writing – original draft:** Marie V. Ozanne.

**Writing – review & editing:** Marie V. Ozanne, Grant D. Brown, Breanna M. Scorza, Kurayi Mahachi, Angela J. Toepp, Christine A. Petersen.

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
