## [Decision Letter · Decision Letter 0]

5 Nov 2021

Dear Dr. Ozanne,

Thank you very much for submitting your manuscript "Bayesian latent class models for identifying canine visceral leishmaniosis using diagnostic tests in the absence of a gold standard" for consideration at PLOS Neglected Tropical Diseases. As with all papers reviewed by the journal, your manuscript was reviewed by members of the editorial board and by several independent reviewers. In light of the reviews (below this email), we would like to invite the resubmission of a significantly-revised version that takes into account the reviewers' comments. 

We cannot make any decision about publication until we have seen the revised manuscript and your response to the reviewers' comments. Your revised manuscript is also likely to be sent to reviewers for further evaluation.

Sincerely,

Richard Reithinger

Associate Editor

Charles Jaffe

Deputy Editor

Reviewer's Responses to Questions

**Key Review Criteria Required for Acceptance?**

**Methods**

-Are the objectives of the study clearly articulated with a clear testable hypothesis stated?

-Is the study design appropriate to address the stated objectives?

-Is the population clearly described and appropriate for the hypothesis being tested?

-Is the sample size sufficient to ensure adequate power to address the hypothesis being tested?

-Were correct statistical analysis used to support conclusions?

-Are there concerns about ethical or regulatory requirements being met?

Reviewer #1: The authors used the babesyan test model to associate and detect dogs with possible asymptomatic and symptomatic visceral leishmaniasis. The animals were tested by PCR and serological methods. The number of dogs tested is small and the association using this mathematical model was inconclusive, as the authors themselves present in the discussion. The choice of asymptomatic dogs is not clearly described in the study.

Reviewer #2: Yes, objectives of the study are clearly articulated. Study design and statistical analyses are appropriate.

**Results**

-Does the analysis presented match the analysis plan?

-Are the results clearly and completely presented?

-Are the figures (Tables, Images) of sufficient quality for clarity?

Reviewer #1: The description of the methods is not clear. The methodology of PCR and serological testing should be better and more clearly described. Most importantly are such methods reproducible?

Reviewer #2: Results are clearly and completely presented. Figures (Tables, Images) are of sufficient quality for clarity.

**Conclusions**

-Are the conclusions supported by the data presented?

-Are the limitations of analysis clearly described?

-Do the authors discuss how these data can be helpful to advance our understanding of the topic under study?

-Is public health relevance addressed?

Reviewer #1: Sample banks of positive and negative dogs for visceral leishmaniasis were used. I ask what is understood in the text. The limitations of the methodology used are discussed by the authors and are not few. What unfortunately does not prove the fidelity of the study. Anyway, this method does not seem feasible in public health. The detection of asymptomatic dogs is a great challenge, especially in endemic areas of developing countries. Although the authors' attempt was praised, the study did not prove to be a possible and simple methodology to apply in such areas. The ideal would be to find a reliable serological test to determine the possibility of an asymptomatic dog a posteriori to develop disease and transmit it. Finally, the authors did not study whether asymptomatic dogs are likely to be reservoirs.

Reviewer #2: Conclusions RE supported by the data presented. Limitations of analysis are clearly described.

**Editorial and Data Presentation Modifications?**

Reviewer #1: I suggest further review of the article. Let this be clearer especially in the detection of asymptomatic dogs, especially in the description of the methodology, in the separate use of PCR and the technique used to observe and describe the results. Finally, the clinical follow-up of asymptomatic dogs compared to the results obtained could give greater credibility to the results. We have no follow up on this dog population.

Reviewer #2: Mathematical notation needs to be improved:

- In equation (1), the probability of the Bernoulli distribution is denoted by \\delta_i. However, in the next line, this is shown as \\delta_{ij}. Authors need to clarify and reconcile the notation differences (different indices).

- x_i's and the design matrix are not clearly defined.

**Summary and General Comments**

Reviewer #1: Interesting study in the search for answers to two canine populations using PCR and serological technique. I suggest the authors complement the study as described above and improve the description of the methodology. It is important to validate a reliable and especially simple method for use in a neglected disease. As it is presented, I do not recommend publication and praise the authors, but I do advise improving it with revision.

Reviewer #2: Authors discuss an interesting application and successfully take advantage of the Bayesian hierarchical modeling framework to combine diagnostic data from two different tests (PCR and DPP) to inform the true status of the disease presented by the latent process (D). In doing so they consider two sets of outcomes: dichotomized PCR and DPP, and dichotomized PCR and DPP reader outcomes and show the benefits of using the DPP reader data (as opposed to the common practice of using dichotomized DPP outcomes). 

Authors also take advantage of the Bayesian framework by considering informative priors based on sensitivity and specificity of the tests.

Authors do a good job of discussing the limitations of their case study. Mainly, the fact that majority of their cases have negative PCR and DPP outcomes and the number of positive PCR outcomes is very small. Authors may address the limitations of the data and in order to show the merits of their approach by conducting simulation studies however, I do realize that such a simulation is not trivial.

Mathematical notation needs to be improved:

- In equation (1), the probability of the Bernoulli distribution is denoted by \\delta_i. However, in the next line, this is shown as \\delta_{ij}. Authors need to clarify and reconcile the notation differences (different indices).

- x_i's and the design matrix are not clearly defined.

PLOS authors have the option to publish the peer review history of their article (what does this mean?). If published, this will include your full peer review and any attached files.

Reviewer #1: Yes: Prof. Valdir Sabbaga Amato, University of São Paulo, Brazil

Reviewer #2: No
---

## [Editor Report · Decision Letter 1]

6 Feb 2022

Dear Dr. Ozanne,

We are pleased to inform you that your manuscript 'Bayesian latent class models for identifying canine visceral leishmaniosis using diagnostic tests in the absence of a gold standard' has been provisionally accepted for publication in PLOS Neglected Tropical Diseases.

Best regards,

Richard Reithinger

Associate Editor

Charles Jaffe

Deputy Editor

---

## [Editor Report · Acceptance letter]

10 Mar 2022

Dear Dr. Ozanne,

We are delighted to inform you that your manuscript, "Bayesian latent class models for identifying canine visceral leishmaniosis using diagnostic tests in the absence of a gold standard," has been formally accepted for publication in PLOS Neglected Tropical Diseases.

Best regards,

Shaden Kamhawi

co-Editor-in-Chief

Paul Brindley

co-Editor-in-Chief
